**Data Availability Statement:** Data available at https://www.datosabiertos.gob.pe/group/datos-abiertos-de-covid-19.

**Funding:** This manuscript has no funding.

# Sex differences in the incidence, mortality, and fatality of COVID-19 in Peru

**Max Carlos Ramírez-Soto**[1,2]*, **Hugo Arroyo-Hernández**[3], **Gutia Ortega-Cáceres**[4]

**1** School of Public Health and Administration, Universidad Peruana Cayetano Heredia, Lima, Peru, **2** Facultad de Ciencias de la Salud, Universidad Tecnológica del Peru, Lima, Peru, **3** Oficina General de Información y Sistemas, Instituto Nacional de Salud, Lima, Peru, **4** Escuela de Posgrado, Universidad Ricardo Palma, Lima, Peru

* maxcrs22@gmail.com, max.ramirez@upch.pe

## Abstract

### Background

There is a worrying lack of epidemiological data on the sex differential in COVID-19 infection and death rates between the regions of Peru.

### Methods

Using cases and death data from the national population-based surveillance system of Peru, we estimated incidence, mortality and fatality, stratified by sex, age and geographic distribution (per 100,000 habitants) from March 16 to November 27, 2020. At the same time, we calculated the risk of COVID-19 death.

### Results

During the study period, 961894 cases and 35913 deaths were reported in Peru. Men had a twofold higher risk of COVID-19 death within the overall population of Peru (odds ratio (OR), 2.11; confidence interval (CI) 95%; 2.06–2.16; *p*<0.00001), as well as 20 regions of Peru, compared to women (*p*<0.05). There were variations in incidence, mortality and fatality rates stratified by sex, age, and region. The incidence rate was higher among men than among women (3079 vs. 2819 per 100,000 habitants, respectively). The mortality rate was two times higher in males than in females (153 vs. 68 per 100,000 habitants, respectively). The mortality rates increased with age, and were high in men 60 years of age or older. The fatality rate was two times higher in men than in women (4.96% vs. 2.41%, respectively), and was high in men 50 years of age or older.

### Conclusions

These findings show the higher incidence, mortality and fatality rates among men than among women from Peru. These rates vary widely by region, and men are at greater risk of COVID-19 death. In addition, the mortality and fatality rates increased with age, and were most predominant in men 50 years of age or older.

**Competing interests:** We have no conflicts of interest.

## Introduction

While males and females have the same susceptibility to COVID-19 infection, global data have shown higher mortality and fatality rates among men than among women [1]. In addition, most countries with available data have shown higher infection, mortality and fatality rates in males than in females, and these rates varied from country to country and between regions [1–5]. Furthermore, there is an increased risk of death for both sexes with the advancement of age, but at all ages above 30 years, males show a significantly higher risk of death than females [2, 3]. In several countries, the male/female ratio as regards death is above 1, and in some countries, such as Albania, Costa Rica, Thailand and the Netherlands, this ratio is even higher than 2 [2].

The first SARS-CoV-2 infection in Peru was identified in March 2020, which was followed by transmission into the community. Peru is currently still being affected by the SARS-CoV-2 pandemic, with more than 1 million cases and more 35000 deaths reported as of February 14, 2021 [6]. In Peru, the COVID-19 mortality rate is 11.3 per 10,000 habitants, and the fatality rate is 3.54% [6]. One study in May 2020 reported fatality rates in men and women of 10.8 and 6.5%, respectively, for individuals older than 70 years of age who contracted COVID-19 [7]. Despite these findings, data on the effects of sex and age differentials on incidence, mortality and fatality rates for COVID-19, and these rates' associations between regions of Peru, are not available. These estimates are important for refining estimates of infection and transmission via different regions' health profiles. Furthermore, this analysis could help to significantly improve our knowledge, and provide insights into COVID-19 prevention and control in Peru.

For these reasons, in order to describe the differences in SARS-CoV-2 infection and death rates among men and women from Peru, we estimated the incidence, mortality and fatality from COVID-19, stratified by sex, age, and geographic distribution.

## Materials and methods

### Study design

For this study, we obtained case and death data for COVID-19 from the National Open Data Platform, Presidencia del Consejo de Ministros, Peru (https://www.datosabiertos.gob.pe/group/datos-abiertos-de-covid-19), which collects daily information on COVID-19 cases and deaths in Peru. Peru is a multicultural and multilingual country that is divided into 25 regions. The total population of Peru is 31 million inhabitants, according to the National Institute of Statistics and Informatics (INEI). In Peru, the Centro Nacional de Epidemiología, Prevención y Control de Enfermedades (CDC-Peru) maintains the national system for the surveillance of significant diseases, including COVID-19. Starting on 5 March 2020, the CDC-Peru mandated the immediate reporting of all COVID-19 cases. Information that is collected by healthcare centers is directed to the national surveillance system. These data are then published daily in the National Open Data Platform, Peru. COVID-19 cases include all those patients who have been reported to the CDC-Peru with and without symptoms of COVID-19 and laboratory confirmation. Laboratory-confirmed COVID-19 cases are defined as those with a positive result for COVID-19 from an RT-PCR assay or immunochromatographic test. The study period was March 16 to November 27, 2020.

In the National Open Data Platform, data on cases of and deaths from COVID-19 include basic demographic information, such as age, sex, region and date of notification. Information on ethnicity, symptoms and comorbidities is not available in the data on cases and deaths by COVID-19.

All data were collected as part of the routine surveillance of the National Open Data Platform, Peru. All data were fully anonymized before you accessed them. Therefore, the study was exempt from review by an ethics board.

## Statistical analysis

COVID-19 cumulative incidence, mortality and fatality estimates were calculated by regions, and stratified by sex and age group. We used the numbers of COVID-19 cases and deaths along with population estimates to calculate the cumulative incidence and mortality of COVID-19 in the population, according to age and sex, per 100,000 people. The population estimates for Peru in 2020 were obtained from the National Institute of Statistics and Informatics (INEI). We also calculated the differences in the absolute numbers of cases in each region, stratified by sex; for this, we assumed that all the estimated populations of Peru in 2020 were exposed to COVID-19. Finally, we calculated the association between sex and absolute numbers of COVID-19 deaths. Differences in absolute numbers of cases, and the associations between sex and absolute numbers of COVID-19 deaths, were determined using a $\chi 2$ test and odds ratio (OR). Stata Software version 9.4 (SAS Institute) was used for data analyses.

## Results

### COVID-19 cases and deaths

A total of 961,894 cases with laboratory confirmation from March 16 through to November 27, 2020 were include in the study. Of these, 748,229 (77.8%) and 213,665 (22.2%) cases were confirmed by immunochromatographic tests and RT-PCR, respectively. Of all the cases, 498,568 (51.8%) were men. During this period, 35,913 deaths associated with COVID-19 were reported in Peru. Of these, 24,730 (68.9%) were men. Most of the cases and deaths, for both men and women, were reported in the Lima region (Table 1).

On average, the COVID-19 infection rate was significantly higher in men ($p<0.00001$) compared to women, and was higher in 19 regions of Peru (Table 1). Men had a twofold higher risk of COVID-19 death within the overall population of Peru (OR, 2.11; CI 95%; 2.06–2.16; $p<0.00001$), as well as in 20 regions of Peru, compared to women. In the Amazonas region, men had an almost fourfold higher risk of COVID-19 death compared to women (OR, 3.95; CI 95%; 2.92–5.34; $p<0.00001$) (Table 1).

### Incidence rate

There were variations in estimated incidence according to sex and age (Fig 1). Although the incidence of COVID-19 overall was higher among men than among women (3079 vs. 2819 per 100,000 habitants, respectively), the incidence was higher among females in 13 regions of Peru (Fig 1A). The highest incidence rates of COVID-19 among women (8419 per 100,000 women) and men (7494 per 100,000 male) were reported in Moquegua, followed by the Madre de Dios and Lima regions. The incidence rates of COVID-19 in the overall population of Peru increased with age in both men and women (Fig 1B and 1C). The incidence rates were highest in women between the ages of 40 and 49 years (12,592 per 100,000 women) (Fig 1B), and in men in the age group of ≥80 years (12,571 per 100,000 men), in the Moquegua and Amazonas regions, respectively (Fig 1C).

### Mortality rate

Overall, the mortality rate associated with COVID-19 was two times higher in males than in females (153 vs. 68 per 100,000 habitants, respectively) (Fig 2A), in all the regions of Peru. The

**Table 1.** Differences in the cases absolute numbers and risk of COVID-19 deaths, stratified by sex, and geographic distribution, Peru, 2020.

| Region | Population | | COVID-19 cases | | | COVID-19 deaths | | | Risk of COVID-19 death |
|---|---|---|---|---|---|---|---|---|---|
| | Male, No | Female, No | Male, No | Female, No | p-value* | Male, No | Female, No | p-value* | OR (95% CI) |
| Overall | 16,190,895 | 16,435,053 | 498568 | 463326 | 0.00001 | 24730 | 11183 | 0.00001 | 2.11 (2.06–2.16) |
| Amazonas | 219,801 | 207,005 | 8382 | 9566 | 0.00001 | 187 | 55 | 0.00001 | 3.95 (2.92–5.34) |
| Áncash | 594,832 | 585,806 | 14204 | 14426 | 0.0083 | 977 | 450 | 0.00001 | 2.29 (2.05–2.57) |
| Apurímac | 220,370 | 210,366 | 3233 | 3239 | 0.0501 | 80 | 56 | 0.03650 | 1.44 (1.02–2.04) |
| Arequipa | 735,707 | 761,731 | 24010 | 22289 | 0.00001 | 1064 | 469 | 0.00001 | 2.16 (1.93–2.41) |
| Ayacucho | 341,951 | 326,262 | 7139 | 7371 | 0.00001 | 246 | 105 | 0.00001 | 2.47 (1.96–3.11) |
| Cajamarca | 727,265 | 726,446 | 11973 | 11835 | 0.415 | 378 | 182 | 0.00001 | 2.09 (1.75–2.50) |
| Callao | 550,046 | 579,808 | 22025 | 18687 | 0.00001 | 1285 | 617 | 0.00001 | 1.81 (1.64–2.00) |
| Cusco | 686,543 | 670,532 | 12299 | 11675 | 0.026 | 334 | 157 | 0.00001 | 2.05 (1.69–2.48) |
| Huancavelica | 184,121 | 181,196 | 3746 | 3905 | 0.0109 | 93 | 40 | 0.00001 | 2.46 (1.69–3.57) |
| Huánuco | 384,345 | 375,922 | 8946 | 10134 | 0.00001 | 278 | 172 | 0.00001 | 1.86 (1.53–2.25) |
| Ica | 488,836 | 486,346 | 15083 | 15850 | 0.00001 | 1140 | 585 | 0.00001 | 2.13 (1.93–2.36) |
| Junín | 678,494 | 682,973 | 12637 | 12902 | 0.252 | 613 | 286 | 0.00001 | 2.25 (1.95–2.59) |
| La Libertad | 1,000,002 | 1,016,769 | 17918 | 17454 | 0.00004 | 1612 | 762 | 0.00001 | 2.17 (1.98–2.37) |
| Lambayeque | 638,228 | 672,557 | 15273 | 15567 | 0.003 | 1269 | 581 | 0.00001 | 2.34 (2.11–2.58) |
| Lima | 5,119,560 | 5,508,910 | 232188 | 196840 | 0.00001 | 11262 | 4841 | 0.00001 | 2.02 (1.95–2.09) |
| Loreto | 531,000 | 496,559 | 12022 | 12514 | 0.00001 | 705 | 283 | 0.00001 | 2.69 (2.34–3.09) |
| Madre de Dios | 98,215 | 75,596 | 4607 | 4610 | 0.00001 | 112 | 39 | 0.00001 | 2.92 (2.04–4.21) |
| Moquegua | 102,855 | 89,885 | 7708 | 7567 | 0.00001 | 197 | 90 | 0.00001 | 2.17 (1.69–2.80) |
| Pasco | 140,252 | 131,652 | 3152 | 3088 | 0.0874 | 74 | 50 | 0.03920 | 1.46 (1.02–2.09) |
| Piura | 1,030,975 | 1,016,979 | 20532 | 20242 | 0.954 | 1386 | 712 | 0.00001 | 1.99 (1.81–2.18) |
| Puno | 611,616 | 626,381 | 9454 | 9004 | 0.00001 | 250 | 120 | 0.00001 | 2.01 (1.61–2.50) |
| San Martín | 474,458 | 425,190 | 11313 | 12721 | 0.00001 | 531 | 236 | 0.00001 | 2.60 (2.23–3.04) |
| Tacna | 188,152 | 182,822 | 7089 | 6938 | 0.663 | 178 | 73 | 0.00001 | 2.42 (1.84–3.19) |
| Tumbes | 135,675 | 115,846 | 4441 | 4619 | 0.00001 | 221 | 111 | 0.00001 | 2.13 (1.69–2.68) |
| Ucayali | 307,596 | 281,514 | 9194 | 10283 | 0.00001 | 258 | 111 | 0.00001 | 2.65 (2.11–3.31) |

*$\chi$2 test

Abbreviations: OR, odds ratio; CIs, confidence intervals.

highest mortality rates for men and women were reported in the Callao region (234 vs. 106 per 100,000 habitants), followed by Ica, Lima and Moquegua (Fig 2A). On average, the mortality rates increased with age, and the mortality rates in men 60 years of age or older, as well as in the 25 regions of Peru we assessed, were high (Fig 2B) compared with women 60 years of age or older (Fig 2C). The highest mortality rates for men and women were reported in the age group of ≥80 years, and were found in the Tumbes (2590 per 100,000 men) and Ica (1288 per 100,000 women) regions, respectively (Fig 2A and 2B).

## Fatality rate

The fatality rate associated with COVID-19 in the overall population of Peru was two times higher in men than in women (4.96% vs. 2.41%, respectively) (Fig 3A). The fatality rate was higher in men in all regions of Peru. The fatality rates in men 50 years of age or older, as well as those in the 25 regions of Peru, were high (Fig 3B), compared with women 50 years of age or older (Fig 3C). The highest fatality rates were reported in men and women in the age group of ≥80 years, and were found in the Loreto (57.31%) and Ica (37.74%) regions, respectively (Fig 3B and 3C).

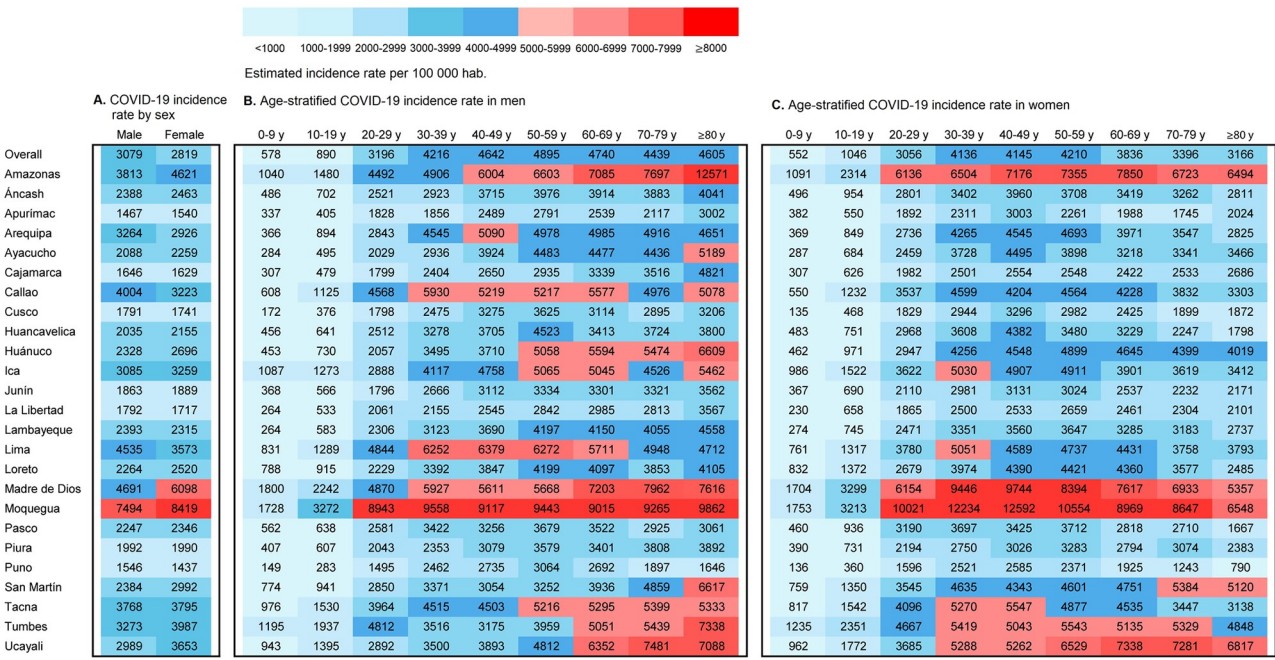

**Fig 1. Sex and age-stratified COVID-19 incidence estimates by region (A, B and C), Peru, 2020.**

## Discussion

In this study, based on the national population surveillance system, we found higher rates of incidence, mortality and fatality among men than among women in Peru. In addition, the

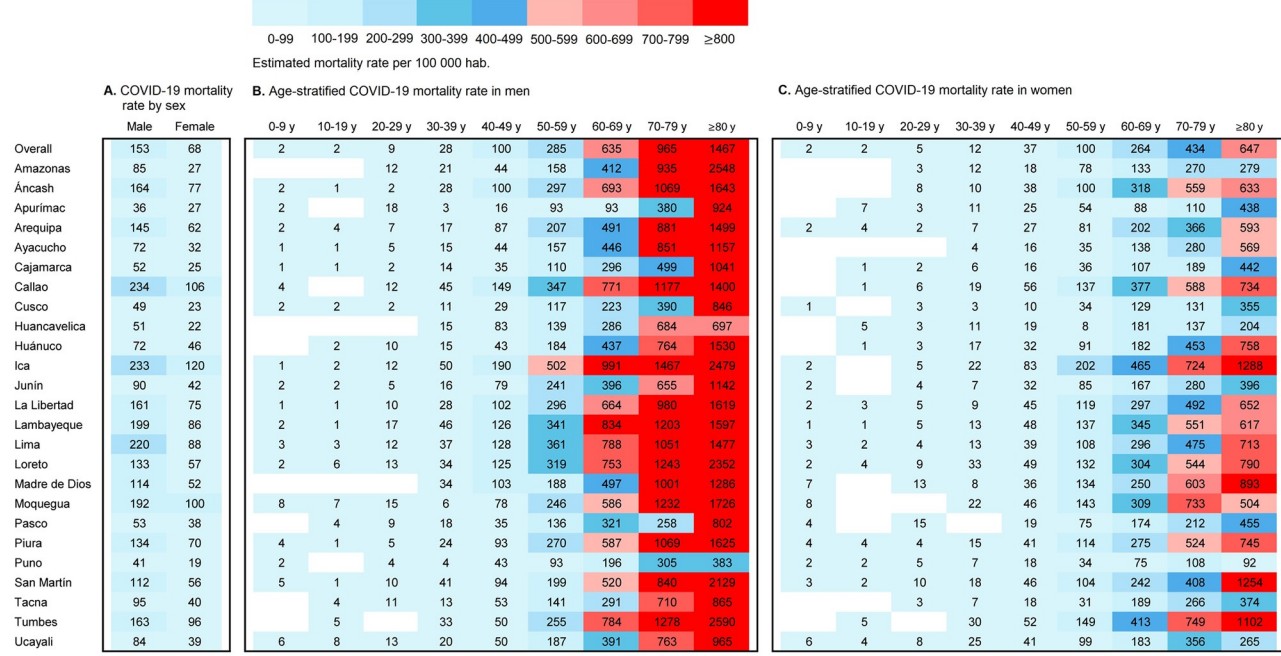

**Fig 2. Sex and age-stratified COVID-19 mortality estimates by region (A, B and C), Peru, 2020.**

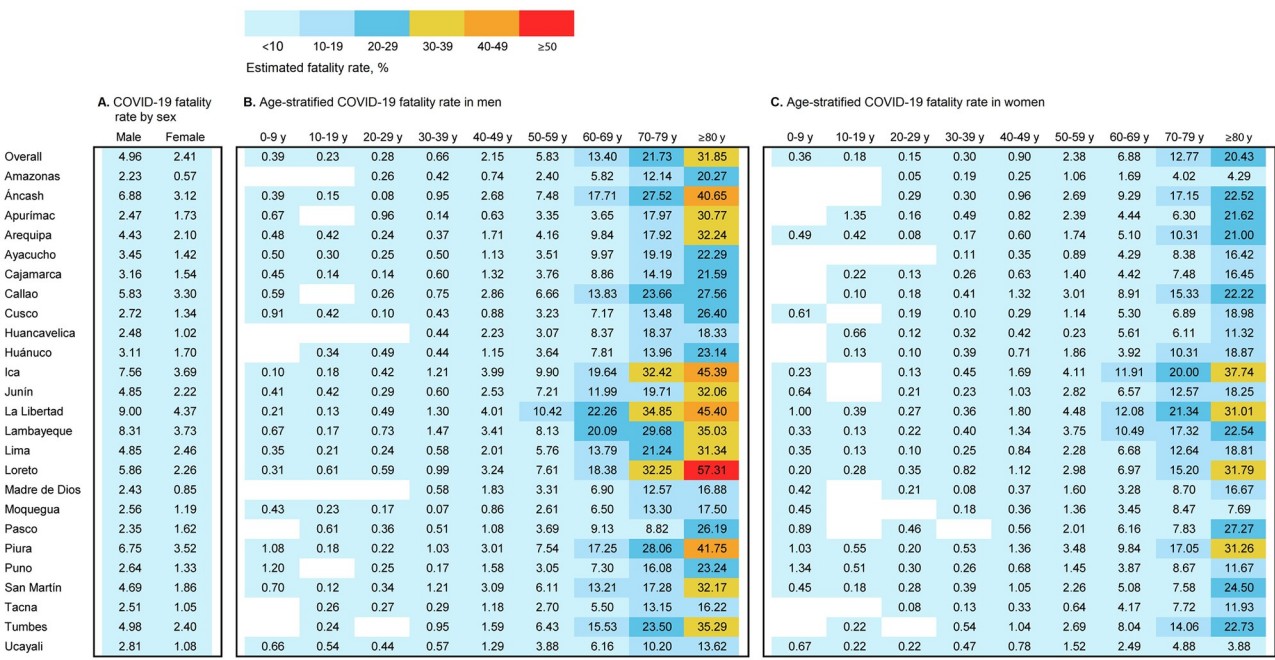

**Fig 3. Sex and age-stratified COVID-19 fatality estimates by region (A, B and C), Peru, 2020.**

mortality and fatality rates increased with age, and were predominant in men 50 years of age or older. To the best of our knowledge, only one study has previously estimated COVID-19 mortality among men and women in Peru (as of May 2020), and this only included totals of 129,148 COVID-19 cases and 7,660 COVID-19 deaths [7], while our study included totals of 961,894 COVID-19 cases and 35,913 COVID-19 deaths. Our principal finding was that men were at a twofold higher risk of COVID-19 death in 20 regions of Peru, and an almost fourfold higher risk of COVID-19 death in 1 region. These findings suggest an increased risk of death from COVID-19 for men throughout almost the whole country. The higher risks of death associated with COVID-19 for men are probably related to cardiovascular diseases, obesity or diabetes, biological or genetic factors, age, and the epidemiological profile in each region, but they can also be explained by the deficiencies of the health system [11–15].

In April 2020, Peru enacted multiple interventions (social isolation, use of masks and hand-washing), including closing businesses and prohibiting gatherings, to prevent SARS-CoV-2 transmission. Despite these nonpharmaceutical interventions, the incidence rates evolved simultaneously, and were higher among men than among women, with the highest rates among women being in the Moquegua and Madre de Dios regions. This can be explained by the lower proportions of women in the populations of some regions (Table 1). In contrast, despite the size of the male population being greater in some regions, the COVID-19 incidence rates registered in some of these regions were higher. Despite these population differences, it should be noted that in six regions of Peru (Ucayali, Huánuco, San Martín, Ica, Junín and Ancash), the highest numbers of cases were registered in women, and five of these regions had higher incidence rates in women than in men. Several reports point to sex differences in COVID-19 resulting from male patients having higher rates of infection. These disparities in sex mainly relate to factors concerning social behaviour and human biology [8]. Among social factors, it is considered that men represent a higher proportion of smokers, and more often exhibit lifestyles that cause the main comorbidities associated with COVID-19. In addition,

men enact cultural practices that put them at greater risk of becoming ill, spreading the infection or seeking less medical attention [8]. The greater susceptibility of men can also be related to their greater amounts of angiotensin-converting enzyme 2 (ACE2) receptors compared to women [9], although further studies are needed to confirm that plasma ACE2 levels can indicate a higher risk for COVID-19 [10]. Besides this, we found disproportionate differences in incidence rates stratified by sex and age. For example, high incidence rates were registered in women aged 20 and 79 years old in 5 regions, whereas the highest incidence rates were recorded among men aged 50 and ≥80 years old in 10 regions of Peru. In addition, the incidence rates for girls and women 10–19 years of age were the highest in all regions of Peru, compared with men. These disproportionate incidence rates stratified by age among men/women are explained in part by social determinants, such as unemployment, educational level, housing and living conditions and population density, in addition to the assessed levels of susceptibility to SARS-CoV-2 infection among the different age groups of men/women in Peru.

In our study, the COVID-19 mortality rates were the highest in men for all regions of Peru. These rates increased with age, and were higher in men 60 years of age or older, compared with women. The mortality rate among adults was higher than that reported by various other studies [1, 3, 15]. These differences in mortality rates associated with older age are probably related to cardiovascular diseases, obesity, or diabetes [11–15], since these are common conditions among older adults [16]. In addition, older adults have elevated rates of COVID-19-associated hospitalization, and the majority of persons hospitalized have underlying medical conditions [17]. In contrast, we found low mortality rates in women [1, 3, 15]. These findings are consistent with the literature. The lower mortality rates in women might be due to their better immune response regulated by estrogen [18, 19]. Women infected with SARS-CoV-2 have been shown to produce more T cells than men [20]. Likewise, it has been reported that in the early phase of the disease, women have relatively higher serum concentrations of IgG antibodies against SARS-CoV-2 [21].

The average fatality rates in Peru are consistent with findings from countries such as Portugal, Germany, Colombia, China, Australia and Bosnia, where the fatality rates are greater for males. In addition, it should be noted that four regions of Peru (La Libertad, Lambayeque, Ica and Piura) show the highest COVID-19 fatality rates. These findings are consistent with the fatality rates from countries such as Mexico, Spain, Ecuador, Switzerland, Romania and the Philippines [2]. The fatality rates also increase with age. As such, in 38 countries reporting sex-disaggregated data on COVID-19 cases and deaths, males above 60 years of age had higher fatality rates than females [2]. In our study, the fatality rates also increased with age, and were highest in men 60 years of age or older, in all the regions of Peru, compared with women, with fatality rates greater than 40% for men ≥80 years old in five regions. Similar to mortality, these fatality rates assessed by age are probably related to cardiovascular diseases, obesity, immune response, socioeconomic and demographic factors, etc [11–15]. To these factors must be added the weaknesses of the Peruvian health system, which at the beginning of the pandemic had only an average of one bed in an intensive care unit for every 100,000 people, as well as a shortage of oxygen and few health professionals.

Sex differences in incidence and mortality from COVID-19 can also be explained for immunological mechanisms, genetic factor, inflammation, and cancer. A recent study found that men had higher levels of IL-8 and IL-18. In addition, women had a more robust CD8 T cell activation, while poor T cell responses were associated with COVID-19 progression in males [20]. Other study revelated that the females may produce larger amounts of neutralizing antibodies, compared with males, especially in the early phase of COVID-19 [21]. The immune mediators were also associated with adverse outcomes of SARS-CoV-2 in men (TNFSF13B, CCL14, CCL23, IL-7, IL-16, and IL-1857) [22]. Because of these immune mechanisms, males

are more likely to develop the cytokine storm associated with poor clinical outcomes. Literature evidence suggests one potential association between COVID-19 and prostate cancer [22]. Interestingly, the complications and mortality from COVID-19 are predominant in men aged 50 years or older, and the risk for prostate cancer increases in men above the age of 50. This association can be explained by the high expression of TMPRSS2 in prostate cancer, and SARS-CoV-2 entry into the host cell [22–25]. Genetic factor as the presence of the double chromosome XX in women, can also play a role in SARS-CoV-2 infection, since the X chromosome contains a large number of genes regulating immunity [26]. In addition to immunological and genetics mechanisms, the estrogen also plays a significant role in immune responses in women [26, 27]. In summary, based on the evidence, the biological sex differences (immunological mechanisms, genetic factor, and inflammation) may affect the pathogenic mechanisms of COVID-19, including the risk for infection, severity, and death.

The study has important limitations. First, although the incidence rates were high, it is likely that these rates were actually higher, since the availability of diagnosis testing was limited early on in the pandemic, and varied nationally. Furthermore, most of the cases were confirmed by rapid tests, the sensitivity limits of which are low compared with molecular tests. Secondly, due to limitations in the epidemiological surveillance system, the mortality and lethality rates may also actually be higher than those estimated in this study, since there have recently been delays in the registration of deaths. Thirdly, this study was an analysis of a secondary database, and although the database is an official data source, there may have been errors or delays in registration that would produce an underreporting of cases. Therefore, these findings could result in a possible bias. Despite these limitations, our findings give us approximations of incidence, mortality and case fatality rates adjusted for age and sex.

## Conclusion

Our findings show higher incidence, mortality and fatality rates among men than among women in Peru. These rates vary widely by region, and men are at greater risk of COVID-19-assocaited death in the overall population in Peru. In addition, the mortality and fatality rates increased with age, and are predominant in men aged 50 years or older. Therefore, actions aimed at improving the surveillance and prevention of infection in the population, and particularly in that portion with comorbidities and the elderly, are necessary in a country where the capacity for hospital response and intensive care is insufficient.

## Author Contributions

**Conceptualization:** Max Carlos Ramírez-Soto.

**Data curation:** Max Carlos Ramírez-Soto, Hugo Arroyo-Hernández, Gutia Ortega-Cáceres.

**Formal analysis:** Max Carlos Ramírez-Soto, Hugo Arroyo-Hernández, Gutia Ortega-Cáceres.

**Investigation:** Max Carlos Ramírez-Soto, Hugo Arroyo-Hernández, Gutia Ortega-Cáceres.

**Methodology:** Max Carlos Ramírez-Soto, Hugo Arroyo-Hernández, Gutia Ortega-Cáceres.

**Software:** Max Carlos Ramírez-Soto, Hugo Arroyo-Hernández.

**Validation:** Max Carlos Ramírez-Soto.

**Writing – original draft:** Max Carlos Ramírez-Soto, Hugo Arroyo-Hernández, Gutia Ortega-Cáceres.

**Writing – review & editing:** Max Carlos Ramírez-Soto, Hugo Arroyo-Hernández, Gutia Ortega-Cáceres.

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
