## [Decision Letter · Decision Letter 0]

13 Apr 2021

PONE-D-21-06294

Sex differences in the incidence, mortality, and fatality of COVID-19 in Peru

PLOS ONE

Dear Dr. Ramírez-Soto,

Thank you for submitting your manuscript to PLOS ONE. After careful consideration, we feel that it has merit but does not fully meet PLOS ONE’s publication criteria as it currently stands. Therefore, we invite you to submit a revised version of the manuscript that addresses the points raised during the review process.

The reviews are relatively concordant.  Please respond to their comments on a point-by-point basis and revised the manuscript accordingly.  For your response to Reviewer 3, note that novelty, or perceived significance, is not a criterion for acceptance at PLOS ONE.

We look forward to receiving your revised manuscript.

Kind regards,

Jeffrey Shaman

Academic Editor

PLOS ONE

Journal Requirements:

2. In your ethics statement in the manuscript and in the online submission form, please provide additional information about the patient records used in your retrospective study. Specifically, please ensure that you have discussed whether all data were fully anonymized before you accessed them.

"We have no conflicts of interest"

"This manuscript is not funding"

Reviewers' comments:

Reviewer's Responses to Questions

**Comments to the Author**

1. Is the manuscript technically sound, and do the data support the conclusions?

Reviewer #1: Yes

Reviewer #2: Partly

Reviewer #3: Yes

2. Has the statistical analysis been performed appropriately and rigorously? 

Reviewer #1: Yes

Reviewer #2: Yes

Reviewer #3: Yes

3. Have the authors made all data underlying the findings in their manuscript fully available?

Reviewer #1: Yes

Reviewer #2: Yes

Reviewer #3: Yes

4. Is the manuscript presented in an intelligible fashion and written in standard English?

Reviewer #1: Yes

Reviewer #2: Yes

Reviewer #3: Yes

5. Review Comments to the Author

Reviewer #1: This study highlights the COVID-19 incidence and mortality among the Peruvian population. The authors show that the incidence as well as mortality rate is high for males compared to females. This is a general trend in most of the countries. This study does not highlight new findings; however, as the authors have indicated, this study could be important for monitoring and introducing infection control measures in Peru. The manuscript is well written and the authors have presented the data comprehensively. Hence, this manuscript can be accepted in the current form.

Reviewer #2: The present study shows higher COVID-19 incidence and related mortality among men than women in Peru. In addition to showing that men are at greater risk of COVID-19 death, the authors also shows that fatality rates increased with age, and that were mainly predominant in men with 50 years of age or older. The present study lacks in novelty, since these findings are only consistent with a very large number of published reports from different European countries, from different Asian countries, including China, and from the USA. The only advantage that I see from this study is the geographical area of assessment, beeing a research study performed in Peru, since COVID-19 reports from South America are still too few compared to the other continents.

Major point

The authors need to add a paragraph in the Discussion on the potential mechanisms involved in the higher COVID-19 incidence and related mortality among men than women. You can also cite and use as a reference the following very recent publication:

Gender differences in the battle against COVID-19: Impact of genetics, comorbidities, inflammation and lifestyle on differences in outcomes. Int J Clin Pract. 2021 Feb;75(2):e13666.

Reviewer #3: The author report higher incidence, mortality and fatality rates among men than among women from Peru. Men are at greater risk of COVID-19 death. The mortality rates increased with age, and were most predominant in men 50 years of age or older.

The manuscript is well written and technically sound. However, for a global audience the manuscript lacks novelty. The gender and age as risk factors for COVID-19 are already well described.

6. PLOS authors have the option to publish the peer review history of their article (what does this mean?). If published, this will include your full peer review and any attached files.

Reviewer #1: No

Reviewer #2: No

Reviewer #3: No

---

## [Author Response · Author response to Decision Letter 0]

26 Apr 2021

Response-to-reviewers: Manuscript PONE-D-21-06294

We thank the Reviewers for their comments and constructive criticism, we believe that the quality of our manuscript has been significantly improved. We have revised our paper in a point-by-point manner. Modifications are in yellow text. 

Reviewer #1: This study highlights the COVID-19 incidence and mortality among the Peruvian population. The authors show that the incidence as well as mortality rate is high for males compared to females. This is a general trend in most of the countries. This study does not highlight new findings; however, as the authors have indicated, this study could be important for monitoring and introducing infection control measures in Peru. The manuscript is well written and the authors have presented the data comprehensively. Hence, this manuscript can be accepted in the current form.

Response: Thank you for your comment.

Reviewer #2: The present study shows higher COVID-19 incidence and related mortality among men than women in Peru. In addition to showing that men are at greater risk of COVID-19 death, the authors also shows that fatality rates increased with age, and that were mainly predominant in men with 50 years of age or older. The present study lacks in novelty, since these findings are only consistent with a very large number of published reports from different European countries, from different Asian countries, including China, and from the USA. The only advantage that I see from this study is the geographical area of assessment, beeing a research study performed in Peru, since COVID-19 reports from South America are still too few compared to the other continents.

The authors need to add a paragraph in the Discussion on the potential mechanisms involved in the higher COVID-19 incidence and related mortality among men than women. You can also cite and use as a reference the following very recent publication: Gender differences in the battle against COVID-19: Impact of genetics, comorbidities, inflammation and lifestyle on differences in outcomes. Int J Clin Pract. 2021 Feb;75(2):e13666.

Response: Thank you for your comment. We have included a paragraph in the Discussion on the potential mechanisms involved in the higher COVID-19 incidence and related mortality among men than women (lines, 228-247). We have also cited and use as a reference the publication: Gender differences in the battle against COVID-19: Impact of genetics, comorbidities, inflammation and lifestyle on differences in outcomes. Int J Clin Pract. 2021 Feb;75(2):e13666.

Reviewer #3: The author report higher incidence, mortality and fatality rates among men than among women from Peru. Men are at greater risk of COVID-19 death. The mortality rates increased with age, and were most predominant in men 50 years of age or older.

The manuscript is well written and technically sound. However, for a global audience the manuscript lacks novelty. The gender and age as risk factors for COVID-19 are already well described.

Response: Thank you for your comment. While the gender and age as risk factors for COVID-19 are already well described in different European countries, including China, and from the USA, COVID-19 reports in South America and Peru are still too few. Therefore, actions aimed at improving the surveillance and prevention of infection in the population, are necessary in a country where the capacity for hospital response and intensive care is insufficient. We also understand that novelty is not a criterion for acceptance at PLOS ONE.

---

## [Decision Letter · Decision Letter 1]

31 May 2021

Sex differences in the incidence, mortality, and fatality of COVID-19 in Peru

PONE-D-21-06294R1

Dear Dr. Ramírez-Soto,

We’re pleased to inform you that your manuscript has been judged scientifically suitable for publication and will be formally accepted for publication once it meets all outstanding technical requirements.

Kind regards,

Jeffrey Shaman

Academic Editor

PLOS ONE

Additional Editor Comments (optional):

Reviewers' comments:

Reviewer's Responses to Questions

**Comments to the Author**

1. If the authors have adequately addressed your comments raised in a previous round of review and you feel that this manuscript is now acceptable for publication, you may indicate that here to bypass the “Comments to the Author” section, enter your conflict of interest statement in the “Confidential to Editor” section, and submit your "Accept" recommendation.

Reviewer #2: All comments have been addressed

2. Is the manuscript technically sound, and do the data support the conclusions?

Reviewer #2: Yes

3. Has the statistical analysis been performed appropriately and rigorously? 

Reviewer #2: Yes

4. Have the authors made all data underlying the findings in their manuscript fully available?

Reviewer #2: Yes

5. Is the manuscript presented in an intelligible fashion and written in standard English?

Reviewer #2: Yes

6. Review Comments to the Author

Reviewer #2: I have no further comments. The authors have addressed all my previous comments.

The article can be published.

7. PLOS authors have the option to publish the peer review history of their article (what does this mean?). If published, this will include your full peer review and any attached files.

Reviewer #2: No

---

## [Editor Report · Acceptance letter]

4 Jun 2021

PONE-D-21-06294R1 

Sex differences in the incidence, mortality, and fatality of COVID-19 in Peru 

Dear Dr. Ramírez-Soto:

I'm pleased to inform you that your manuscript has been deemed suitable for publication in PLOS ONE. Congratulations! Your manuscript is now with our production department. 

Kind regards, 

on behalf of

Prof. Jeffrey Shaman 

Academic Editor

PLOS ONE